# Visualising reaction complexes in amine-based unloaded and $CO_2$-loaded carbon capture solutions

Harrison Laurent [1], Daniel Sault[2], Thomas F. Headen [3], Terri-Louise Hughes [3], James E. Wheatley[4], Christopher M. Rayner [2,4] & Lorna Dougan [1,5] ✉

In power generation and industries where $CO_2$ emissions are unavoidable, carbon capture, utilisation, and storage is an important tool to offset climate change. Many carbon capture agents are blends of aqueous amines, which absorb $CO_2$ and are then thermally regenerated. The physical interactions between solutes play a crucial role in their reactivity and energy requirements for regeneration. Atomically resolved, experimentally derived information about the structure of these solutions, however, has yet to be reported. In this work, we report the structure of two model carbon capture solvents, aqueous sodium and potassium glycinate, in the unloaded and $CO_2$-loaded state by performing structural refinement on H/D isotopically varied neutron diffraction data. This allows us to quantify the structure, frequency, and EPSR-derived pair interaction energetic stability of intermolecular interactions present. Such methodology can be readily applied to other carbon capture solutions, providing unparalleled insight and facilitating their large-scale modelling and rational design.

Climate change as a result of anthropogenic $CO_2$ emissions is a problem of perpetually increasing importance if our established society is to continue to grow[1,2]. One proposed method to tackle this problem is carbon capture, utilisation and storage (CCUS), where $CO_2$ from waste gas streams is removed and utilised in an industrial process, or geologically stored[3]. The development of CCUS is necessary as mankind is not yet at the point where renewables can be deployed at the scale required to meet the global energy demand[4], and industries such as steel, cement, and glass manufacture all unavoidably emit $CO_2$[5,6]. CCUS also enables negative emission technologies, such as bioenergy with carbon capture and storage (BECCS), offering potentially important mechanisms for offsetting residual emissions, and in the longer term, reducing atmospheric $CO_2$ concentrations[7,8].

Aqueous alkanolamines are the most developed systems for post combustion capture, either as single components, or as blends that have complementary performance benefits[9–11]. Of key importance to

the full-scale deployment of carbon capture solvents is accurate industrial scale modelling of the capture/absorption and solvent regeneration/desorption processes and fundamental molecular understanding of the carbon capture solutions. This cross-length scale understanding relies on quantitative information on the interactions between species formed in the solution phase through $CO_2$ capture. However due to the relatively low interaction energies, dynamic nature, and short interaction lifetimes between molecules in solution[12], accessing this information experimentally is extremely challenging.

An ideal tool to address the complex and dynamic equilibria in amine-based carbon capture research is neutron diffraction. This technique provides an isotopically sensitive, non-damaging, deeply penetrating method to observe atom-atom correlations within aqueous systems[13–16]. It also allows H/D isotope substitution to be exploited due to the large and opposite-in-sign coherent scattering lengths of hydrogen and deuterium[17,18]. Atomically resolved structural

[1]School of Physics and Astronomy, University of Leeds, Leeds, UK. [2]School of Chemistry, University of Leeds, Leeds, UK. [3]ISIS Neutron and Muon Source, Rutherford Appleton Laboratory, Harwell Campus, Didcot, UK. [4]C-Capture Ltd., Harrogate, England. [5]Astbury Centre for Structural Molecular Biology, University of Leeds, Leeds, UK. ✉e-mail: L.Dougan@leeds.ac.uk

information in the solution state can be achieved by simultaneously refining all-atom computer simulations against several isotopically varied datasets until a satisfactory convergence between the real and simulated diffraction data is achieved. This process is known as 'empirical potential structure refinement' (EPSR)[13,19], and has been well-used to determine the structure of aqueous salts[20,21], amino acids[14,22], larger structured biomolecules[23,24], and to quantify the EPSR-derived pair interaction energetic interactions between associated species in solution[20,25,26].

In this work we complete a study of an amino acid salt based carbon capture solvent in both the unloaded and $CO_2$ loaded state using neutron diffraction. When combined with $^1H$ and $^{13}C$ NMR spectroscopy and vapour liquid equilibria measurements, we obtain previously inaccessible information on the key interactions between species before and during amine-based carbon capture, information on how carboxylate salts behave in solution, and the impact they can have on a carbon capture process.

Due to the established environmental health risks and substantial parasitic energy requirements for $CO_2$ release and solvent regeneration following capture of commonly researched carbon capture solutions, such as monoethanolamine[27–29], we chose to investigate aqueous amino acid salts as model carbon capture agents. Amino acid salts[30–33] have been investigated as potential replacements for alkanolamines, with absorption kinetics and cyclic capacity being previously established, alongside likely energy requirements[32–35]. These are prepared by dissolution of equimolar amounts of amino acids and metal hydroxide salts in aqueous solution. They are also excellent substrates for more fundamental studies given their simplicity, availability of isotope variants, structural variety, and offer a simple method of tuning important electrostatic-based interactions by altering the metal carboxylate counterion. In this study we investigate aqueous potassium and sodium glycinate (K glycinate and Na glycinate), chosen as the simplest amino acid to aid in speciation identification and isotope accessibility. Glycine salts have also been demonstrated to strike an important balance between $CO_2$ absorption/desorption properties within amino acid based $CO_2$ capture while being less corrosive toward construction materials[36].

It is well known that amines capture $CO_2$ through producing carbamates and bicarbonate[37–40], however more detailed information would provide greater clarity on key aspects of the reactions involved. The reactions for the case of the glycinate anion are shown in Fig. 1. In the first mechanism (Fig. 1(a)), known as the 'two-step' or 'zwitterionic' mechanism[41,42], glycinate first reacts with $CO_2$ to form an intermediate zwitterionic carbamate, which is then deprotonated by a second glycinate to form glycine in its protonated form. Deprotonation may be by direct interaction, or through a proton hopping Grotthus-like mechanism through the water network[37]. The second proposed mechanism is a single step 'termolecular' route (Fig. 1(b)), where two glycinate ions simultaneously coordinate a $CO_2$ molecule via their amine groups, so nucleophilic attack and proton abstraction occur simultaneously[43]. Similarly, the formation of bicarbonate is proposed to occur through multiple routes via hydration of $CO_2$[39,40], however as this pathway creates an additional species for simulation and potentially unwanted aggregates, it is avoided in this work.

These reaction mechanisms therefore allow us to identify three vital structural features before amine-based $CO_2$ capture:

(1).  The coordination shell around the glycinate amine group including water molecules and metal counterions.
(2).  The strength of water-water hydrogen bonding in the bulk solution, as this has been shown to contribute to the rate of proton hopping[44].
(3).  The frequency of glycinate amine – glycinate amine coordination prior to approach of the $CO_2$ substrate.

It is expected that these reaction routes are equally important for the reverse desorption reaction which allows for the solvent to be regenerated for subsequent reuse[45,46], hence the corresponding structural correlations of the $CO_2$ loaded samples are of equal interest. These are therefore:

(4).  The coordination shell around the glycine carbamate group including water molecules and metal counterions.
(5).  The strength of water-water hydrogen bonding in the bulk solution.
(6).  The frequency of glycine carbamate – glycine zwitterion ammonium coordination.

These structures are outlined in Fig. 2 and numbered in relation to the interactions above. Neutron diffraction is used to examine these crucial structural features and quantify the energetic interactions between species present in amine-based carbon capture solutions, amino acid salts in this case, before and after $CO_2$ loading at unprecedented resolution.

## Results

### Neutron diffraction and structural refinement

The neutron diffraction data on unloaded and $CO_2$ loaded K/Na glycinate concentrations was obtained at 2.17 mol glycinate salt/kg $H_2O$. $^{13}C$ NMR spectroscopy was used to identify and quantify the species

**Fig. 1 | Reactions between CO₂ and glycinate salts.** Reaction of glycinate anion with $CO_2$ forms a carbamate zwitterion pair through either the 'two step' route (**a**) or the 'termolecular' route (**b**), or a bicarbonate zwitterion pair (**c**).

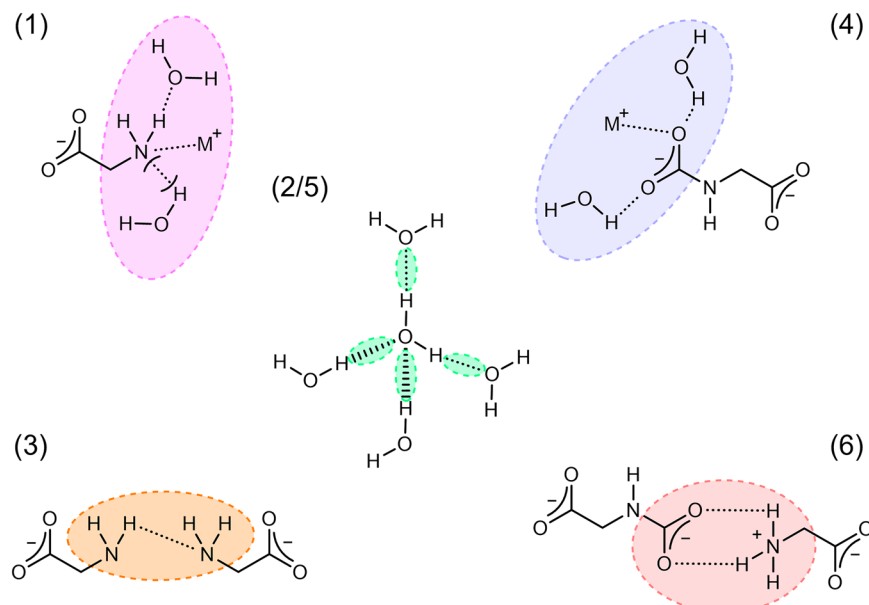

**Fig. 2 | Key structural features of interest relevant to CO$_2$ capture and release in aqueous glycinate salts.** Structural features highlighted by coloured ellipses. These features are: 1) the coordination shell around the unloaded glycinate amine group, 2/5) the bulk water hydrogen bond network pre and post CO$_2$ loading, 3) unloaded glycinate – glycinate pairing via their amine groups, 4) the coordination shell around the loaded glycine carbamate group, 6) loaded glycine carbamate – glycine zwitterion pairing.

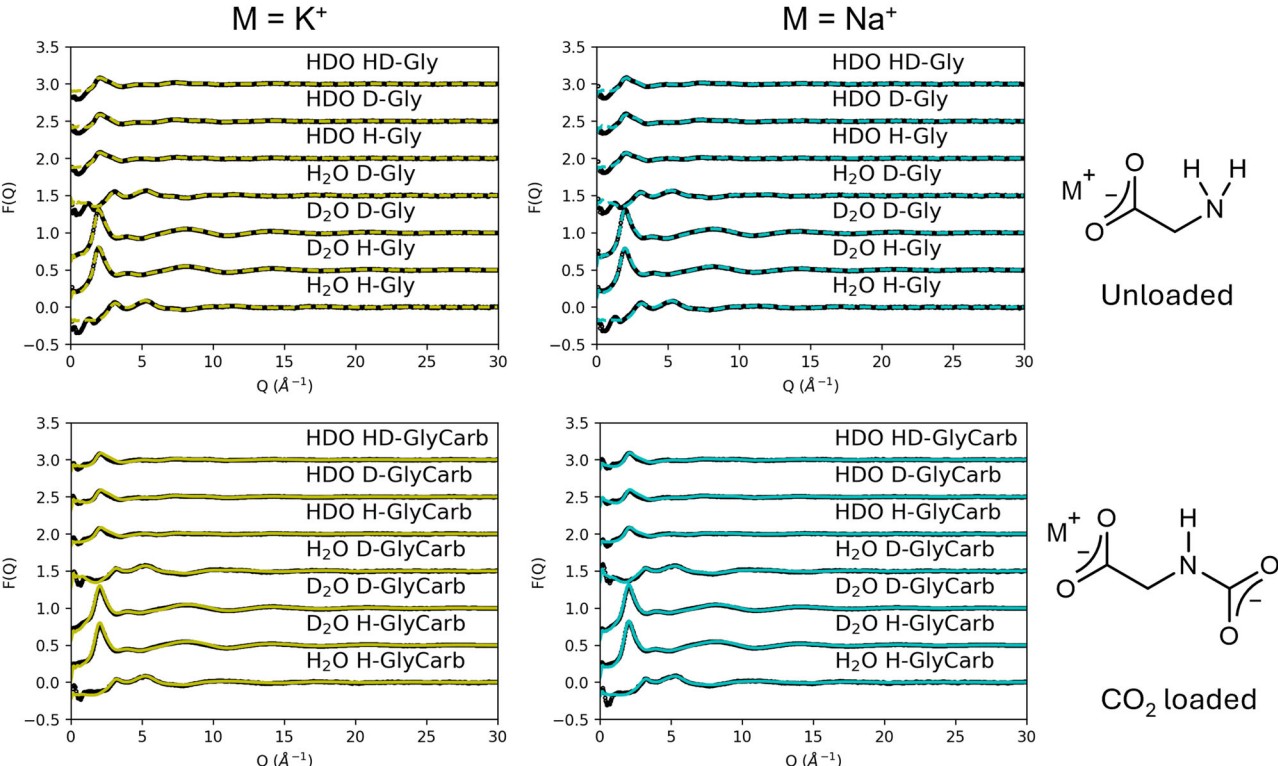

**Fig. 3 | Structural refinement modelling fits to experimental neutron diffraction data.** Experimental data shown as black points for 7 H/D isotope variants of K/Na glycinate with ESPR fits shown as yellow/cyan lines in the unloaded (dashed lines) and CO$_2$ loaded (solid lines) state.

formed through the CO$_2$ reaction process, as shown in supplementary Figs. S1-S3. This allowed us to avoid the generation of bicarbonate ions by limiting the CO$_2$ loading to 0.35 mol equivalents relative to K/Na glycinate (0.76 mol CO$_2$/kg H$_2$O). The final fitted data following structural refinement of experimental data for K/Na glycinate in the loaded and unloaded state are shown in Fig. 3. Here we see high quality

fits are achieved as the yellow/cyan coloured structurally refined model data is almost indistinguishable from the black experimental data, confirming robust modelling of the experimental solutions. Residuals between the fitted and experimental data, as well as fit quality quantification through the 'R factor' are shown in the supplementary Fig. S4 and supplementary equation S1. The direct Fourier

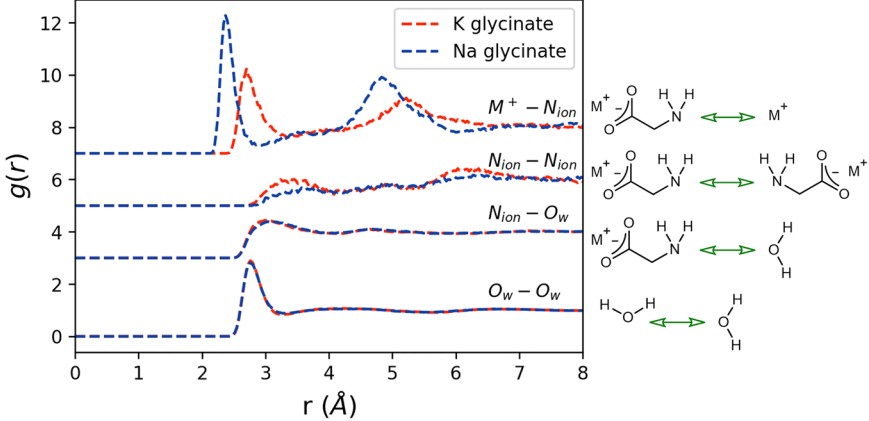

**Fig. 4 | Radial distribution functions for key structural features of interest in aqueous unloaded K/Na glycinate (red dashed/blue dotted lines).** Intermolecular correlations are described using chemical structures, with the particular interatomic correlations of interest shown using green arrows.

transforms of the fitted and experimental data to validate the simulated intramolecular structures are shown in supplementary Fig. S5. These results will now be discussed in detail.

## Unloaded glycinate salts

Interatomic correlations between two species α and β can be described using radial distribution functions ($g_{\alpha\beta}(r)$s), which measure the local density normalised to the bulk density of a particular atom type β relative to a central atom type α as a function of distance. Due to steric hindrance, they therefore begin at 0 at small distances, and decay to unity at large distances, with short range coordination shells appearing as a series of peaks and troughs. A random interatomic structure would therefore occur as a smooth sigmoidal-like growth from 0 to unity. The $g_{\alpha\beta}(r)$s in turn can be used to calculate the number of occurrences of species β around species α, or coordination number, within a minimum and maximum cutoff distance $N_{\alpha\beta}(r_{\min}, r_{\max})$. This is shown in Eq. (1), where $\rho_\beta\rho_\beta$ is the atomic density of atomic species $\beta$.

$$N_{\alpha\beta}(r_{\min}, r_{\max}) = 4\pi\rho_\beta \int_{r_{\min}}^{r_{\max}} r^2 g_{\alpha\beta}(r)dr \qquad (1)$$

The $g_{\alpha\beta}(r)$s corresponding to the structures highlighted in the Introduction section are shown in Fig. 4. Here we observe clear peaks and troughs, indicating a degree of coordination between the considered species.

Structural feature (1), the coordination shell around the central glycinate amine nitrogen ($N_{ion}$), is best discussed in terms of its associations with water molecules, centred around the oxygen ($O_w$), and the metal counterions ($M^+$). The $g_{N_{ion}O_w}(r)$s are very similar in both aqueous K and Na glycinate. A slightly higher peak and slightly lower trough is observed in K glycinate, suggesting a more stable water coordination shell. This more stable coordination shell is reflected in a lower coordination number $N_{N_{ion}O_w}$ for K glycinate, and hence a more expanded local water network. These are calculated to be 7.05 and 7.17 for K and Na glycinate respectively when calculated over a radial distance of 4.08 Å, corresponding to the location of the first minima in the $g_{N_{ion}O_w}(r)$s and hence the width of the first coordination shell. Finally, the average $N_{ion}O_w$ interaction EPSR-derived pair interaction energy $E_{EPSR}$ can be calculated over the first coordination shell using a previously described analysis routine[20,25,26], and is given as $-14.05 \pm 0.07$ and $-13.2 \pm 0.2$ kJ/mol for K and Na glycinate respectively. Interestingly, both values are significantly less stable than the average water–water hydrogen bond EPSR-derived pair interaction energy, given as $-17.71 \pm 0.08$ kJ/mol when calculated previously using the same methodology[25], suggesting significant disruption to water-water interactions in the vicinity of the glycinate amine group.

While the $g_{N_{ion}O_w}(r)$s are observed to be very similar, the $g_{N_{ion}M^+}(r)$s show significant differences between K and Na glycinate. Here the smaller size, and therefore greater charge density, of the $Na^+$ ion relative to the $K^+$ ion allows for closer approach to the amine group, resulting in a shorter first peak distance. This effect has been previously observed when one considers $K^+/Na^+$-water correlations[20,21,47,48], as shown in supplementary Figs. S6 and S7. This also results in a very stable EPSR-derived pair interaction energetic interaction, calculated to be $-318 \pm 1$ and $-367 \pm 1$ kJ/mol for K and Na glycinate respectively. Despite this closer approach and more stable interaction, the $N_{N_{ion}M^+}$ coordination numbers when calculated over 3.15 Å (representing the average location of the first minima in the $g_{N_{ion}M^+}(r)$s) are both equal to 0.14, suggesting equal levels of ion pairing. Taken together, these results show that charge-based interactions are the dominant energetic interaction around the carbon capture agent amine group compared with hydrophobic style entropically driven hydration. Similar effects have been previously observed in neutron diffraction studies of aqueous amino-acids[14,22,49,50]. Accurate quantification of these interactions is vital to understand amine-$CO_2$ reaction mechanisms, as these interactions must be overcome by an approaching $CO_2$ molecule in the two-step reaction mechanism. The clear difference between $N_{ion}M^+$ associations in K glycinate and Na glycinate also provides insight into the physical chemistry where K amino acid salts show enhanced reaction kinetics compared with Na amino acid salts at equal concentrations[32,34,35]. The more stable $N_{ion}Na^+$ interaction contributes to a larger energetic barrier to $CO_2$ absorption. Similar observations have also been made for the hindered approach of urea to peptide backbones by salts[51]. We also validate these previous kinetic studies from the literature using time-resolved vapour liquid equilibrium (VLE) measurements shown in supplementary Fig. S8. It is important at this point to note that EPSR provides an ensemble averaged equilibrium structure, rather than time resolved $CO_2$ uptake data derived through VLE, hence while the two effects are certainly related and logical consistencies clearly exist, one cannot serve as a validation of the other. Unsurprisingly, due to the overall negatively charged nature of the carboxylate group on the glycinate anion, one also clearly observes metal cation – carboxylate interactions. These are both more frequently occurring, with coordination numbers of 0.54 and 0.61 for K glycinate and Na glycinate respectively when calculated over 4.05 Å (representing the average location of the first minima in the $g_{C_{ion}M^+}(r)$s), and more energetically stable, calculated to be $-408 \pm 2$ and $-479 \pm 7$ kJ/mol respectively. While these correlations do not have an appreciable direct impact on the $CO_2$ absorption reaction for glycinate salts, and hence are not discussed here in detail, for the interested reader these structural results and the $g_{C_{ion}M^+}(r)$s are shown in supplementary Fig. S9 and table S1.

**Table 1 | Key structural observations in aqueous K glycinate and Na glycinate determined through neutron diffraction and structural refinement**

| Feature | K glycinate | Na glycinate |
|---|---|---|
| Modal nearest neighbour $N_{ion}O_w$ distance (Å) | 3.02 | 3.08 |
| $N_{N_{ion}O_w}\left(4.08\text{Å}\right)$ | 7.05 | 7.17 |
| $E_{EPSR, N_{ion}O_w}$ (kJ/mol) | −14.05 ± 0.07 | −13.2 ± 0.2 |
| Modal nearest neighbour $N_{ion}M^+$ distance (Å) | 2.71 | 2.38 |
| $N_{N_{ion}M^+}\left(3.15\text{Å}\right)$ | 0.14 | 0.14 |
| $E_{EPSR, N_{ion}M^+}$ (kJ/mol) | −318 ± 1 | −367 ± 1 |
| Modal nearest neighbour $O_wO_w$ distance (Å) | 2.77 | 2.77 |
| $N_{O_wO_w}\left(3.30\text{Å}\right)$ | 3.86 | 3.97 |
| $E_{EPSR, O_wO_w}$ (kJ/mol) | −16.9 ± 0.1 | −16.76 ± 0.09 |
| Modal nearest neighbour $N_{ion}N_{ion}$ distance (Å) | 3.41 | 3.65 |
| $N_{N_{ion}N_{ion}}\left(4.14\text{Å}\right)$ | 0.17 | 0.12 |
| $E_{EPSR, N_{ion}N_{ion}}$ (kJ/mol) | N/A | N/A |

Modal nearest neighbour distances correspond to locations of first peaks in relevant $g_{\alpha\beta}$(r)s. Errors on reported energies correspond to fitting error of location of gaussian peak to calculated EPSR-derived pair interaction energy distributions.

Structural feature (2), the bulk water structure, informs us about the likelihood of proton hopping through the water network, with stronger more linear hydrogen bonds proposed to be more likely to cause hydrogen exchange[44]. This occurs in the deprotonation step of the glycine carbamate zwitterion intermediate in the 'two-step' mechanism. The $g_{O_wO_w}(r)$s in Fig. 4 show a marginally higher first peak and lower trough when comparing K glycinate to Na glycinate, similar to the trends observed for water hydrating the glycinate amine group. This is also consistent with experimental and modelling studies, which describe enhanced water-water hydrogen bonding in solutions of K⁺ ions relative to Na⁺ ions[52–54], owing to the difference in their ionic radii and therefore charge density. The bulk water-water hydrogen bond EPSR-derived pair interaction energies, calculated as described previously[20,25,26] and in the methods section, are measured as −16.9 ± 0.1 and −16.76 ± 0.09 kJ/mol for K and Na glycinate respectively. Using the described cutoff distances means that approximately 24.2% and 25.0% of the total number of available water molecules in the simulations for K-glycinate and Na-glycinate are 'bulk' water molecules respectively. These values both lie within error of one another and are slightly less stable than water-water hydrogen bonds found in pure water, previously calculated to be −17.71 ± 0.08 kJ/mol[25], suggesting that proton hopping in amine-based capture agents is likely to be slower than in pure water[44]. Pulsed field gradient spin echo NMR data presented in the supplementary Fig. S10 also shows that water molecules in K/Na glycinate exhibit slower rotational and translational diffusivity, more so for Na glycinate, suggesting proton hopping enabling structures to occur less frequently.

Finally, structural feature (3), glycinate-glycinate association via amine groups is indeed observed by small peaks and troughs in the $g_{N_{ion}N_{ion}}(r)$s shown in Fig. 4. This occurs despite the overall similar charges of the full molecule, which one may expect to lead to electrostatic repulsion and a relatively featureless $g_{N_{ion}N_{ion}}(r)$. This suggests that the termolecular route can occur even in amine-based carbon capture agents where the capture agents are mutually electrostatically repulsive. It is therefore likely to be more prominent in other neutral amine-based capture agents[14,22,55], such as monoethanolamine[56]. Both the peaks in the $g_{N_{ion}N_{ion}}(r)$ and the $N_{N_{ion}N_{ion}}$ when calculated over 4.14 Å are observed to

be higher in the case of K glycinate compared with Na glycinate, with $N_{N_{ion}N_{ion}}$ calculated to be 0.17 and 0.12 respectively. This difference is likely attributed to the more stable EPSR-derived pair interaction energetic $N_{ion}Na^+$ interaction relative to $N_{ion}K^+$, preventing direct association. This same mechanism can also be used to rationalise the less stable $N_{ion}O_w$ interactions. Unfortunately, this relatively infrequent pairing, combined with their similar charges, makes accurate quantification of their energetic interactions unreliable. The key structural observations of unloaded glycinate solutions made possible through neutron diffraction and structural refinement are finally summarised in Table 1.

## CO₂ loaded glycinate salts

Greater understanding of the CO₂ capture process and its thermo-dynamic reversibility also requires knowledge of the CO₂ loaded state and its environment, which can then be compared with those of the unloaded state. These therefore now primarily centre around the newly formed carbamate group, rather than the amine group *vide supra*. The key structural features are now: (4) The coordination shell around the glycine carbamate group including water molecules and metal counterions. (5) The strength of water-water hydrogen bonding in the bulk solution. (6) The frequency of glycine carbamate – glycine zwitterion amine coordination. The $g(r)$s corresponding to these features are shown in Fig. 5.

Structural feature (4), the coordination shell around the carbamate group $C_{carb}$ is expected to play a key role in decarboxylation, as stable interactions between this group and surrounding molecules will result in a greater required overall energy input. As previously, this is best discussed in terms of its associations with water molecules, centred around the oxygen ($O_w$), the metal counterions ($M^+$), and the zwitterionic glycine, centred around the ammonium nitrogen ($N_{zwit}$). The $g_{C_{carb}O_w}(r)$s are again very similar, with a slightly higher first peak observed in Na glycine carbamate. This results in a higher coordination number when calculated over 4.50 Å in both cases, 10.12 and 10.42 for K glycine carbamate and Na glycine carbamate respectively, however the EPSR-derived pair interaction energetic interactions calculated over the first coordination shell are still more stable for the K glycine carbamate. These are measured to be −29.4 ± 0.2 and −28.3 ± 0.3 kJ/mol for K glycine carbamate and Na glycine carbamate respectively.

It is worth noting at this point that these interactions are significantly more energetically stable than the water – amine interactions quantified in the unloaded state, likely due to the two well solvent exposed, partially negatively charged oxygen atoms in the carbamate group. This allows water molecules to orient themselves to act readily as hydrogen bond donors exclusively. However, in the unloaded amine group as it is modelled here and in previous research[14,55,57], both the partially negatively charged amine nitrogen and partially positively charged amine hydrogen atoms are well solvent exposed, hence optimal orientation of the surrounding water molecules to act as both hydrogen bond donors and acceptors is more challenging. The $C_{carb}M^+$ interactions again occur at shorter distances and are much more energetically stable in Na glycine carbamate compared with K glycine carbamate due to the smaller size and higher charge density of the Na⁺ ion relative to K⁺. These are calculated to be −487 ± 1 and −420 ± 2 kJ/mol respectively. Similar to the $C_{carb}O_w$ EPSR-derived pair interaction energetic interactions compared with the $N_{ion}O_w$ interactions, the $C_{carb}M^+$ interactions are much more stable than the $N_{ion}M^+$ interactions, again likely since the partially negatively charged carbamate oxygen atoms are readily solvent exposed.

This provides an interesting opportunity to compare similar chemical groups within the same molecule, as the newly formed carbamate group is similar to the existing carboxyl group on the glycinate anion. As the newly formed carbamate is bound to the relatively electronegative amine nitrogen, rather than bound to the α-carbon as is the case for the existing carboxyl group, the generated OPLS-AA

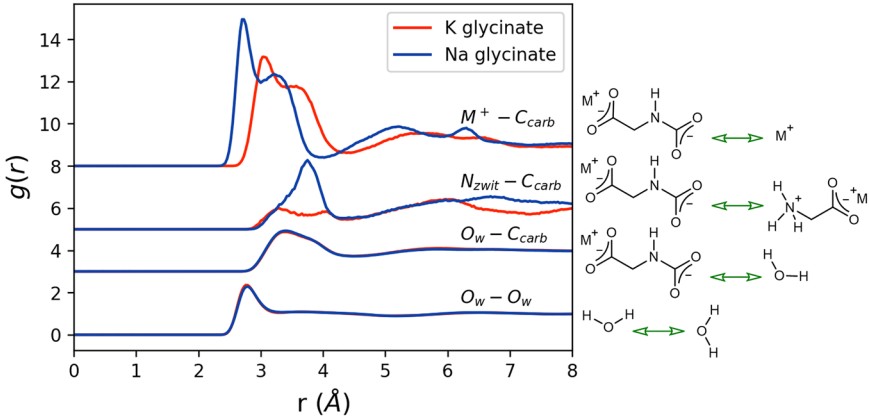

**Fig. 5 | Radial distribution functions for key structural features of interest in aqueous CO₂ loaded K/Na glycinate (red solid/blue dashdotted lines).** Intermolecular correlations are described using chemical structures, with the particular interatomic correlation of interest shown using green arrows.

**Table 2 | Key structural observations in aqueous K glycinate and Na glycinate determined through neutron diffraction and structural refinement**

| Feature | K glycinate | Na glycinate |
|---|---|---|
| Modal nearest neighbour $C_{carb}O_w$ distance (Å) | 3.40 | 3.41 |
| $N_{C_{carb}O_w}\left(4.50\,\text{Å}\right)$ | 10.1 | 10.4 |
| $E_{EPSR,\,C_{carb}O_w}$ (kJ/mol) | −29.4 ± 0.2 | −28.3 ± 0.3 |
| Modal nearest neighbour $C_{carb}M^+$ distance (Å) | 3.06 | 2.72 |
| $N_{C_{carb}M^+}\left(4.20\,\text{Å}\right)$ | 0.74 | 0.7 |
| $E_{EPSR,\,C_{carb}M^+}$ (kJ/mol) | −420 ± 2 | −487 ± 1 |
| Modal nearest neighbour $O_wO_w$ distance (Å) | 2.78 | 2.78 |
| $N_{O_wO_w}\left(3.30\,\text{Å}\right)$ | 3.76 | 3.81 |
| $E_{EPSR,\,O_wO_w}$ (kJ/mol) | −16.04 ± 0.09 | −16.17 ± 0.08 |
| Modal nearest neighbour $C_{carb}N_{zwit}$ distance (Å) | 3.31 | 3.75 |
| $N_{C_{carb}N_{zwit}}\left(4.38\,\text{Å}\right)$ | 0.07 | 0.14 |
| $E_{EPSR,\,C_{carb}N_{zwit}}$ (kJ/mol) | −164.9 ± 0.9 | −106 ± 2 |

Modal nearest neighbour distances correspond to locations of first peaks in relevant $g_{\alpha\beta}(r)$s.
Errors on reported energies correspond to fitting error of location of gaussian peak to calculated EPSR-derived pair interaction energy distributions.

force fields[58] given in the supplementary table S2 predict slightly greater absolute charges. Following refinement of the simulation against the experimental data, we observe a slightly more expanded water structure around the carboxyl group relative to the carbamate group, with the first peak of the $g_{C_{carboxyl}O_w}(r)$ moving outwards, the coordination numbers decreasing, and the EPSR-derived pair interaction energetic interactions becoming less stable. Similar results are also observed for the carboxyl – cation interactions and are summarised in supplementary Fig. S12 and table S3.

Structural feature (5), the bulk water network, is now significantly perturbed upon CO₂ loading. We observe a reduced degree of tetrahedral ordering evidenced by a significant reduction in peak height of the $g_{O_wO_w}(r)$s while maintaining similar coordination numbers. The second peak in the $g_{O_wO_w}(r)$s is also observed to move to shorter distances, suggesting a more compact structure overall. These peak changes are likely explained by a greater abundance of charged species upon CO₂ loading as the added carbamate group is significantly polar, which disrupts the water hydrogen bond network. Certainly, similar

effects are commonly observed with high ionic strength solutions, or increasing pressure[26,59,60]. Unlike previously, the K glycine carbamate bulk water network is observed to be less stable than the Na glycine carbamate, with calculated water-water hydrogen bond EPSR-derived pair interaction energies of −16.04 ± 0.09 and −16.17 ± 0.08 kJ/mol respectively, however these two values lie within error of one another. Using the described cutoff distances means approximately 26.8% of the total number of available water molecules in the simulations for both loaded K-glycinate and Na-glycinate are 'bulk' water molecules.

Finally, structural feature (6), glycine carbamate – glycine zwitterion interactions, are strongly observed through clear peaks in the $g_{C_{carb}N_{zwit}}(r)$s. As these species are now no longer similarly charged, and the partially positively charged amine group can interact energetically favourably with the partially negatively charged carbamate group, the peaks in the $g_{C_{carb}N_{zwit}}(r)$s are much more prominent than those in the $g_{N_{ion}N_{ion}}(r)$s. Here we observe the opposite trend to the unloaded data, that the peaks in the $g_{C_{carb}N_{zwit}}(r)$ are higher in Na glycine carbamate compared with K glycine carbamate. Correspondingly, the coordination number is higher when calculated over 4.38 Å, given as 0.07 and 0.14 respectively. However, despite the lower coordination number, the EPSR-derived pair interaction energetic interaction is calculated as previously to be more stable in K glycine carbamate than Na glycine carbamate, given as −164.9 ± 0.9 and −106 ± 2 kJ/mol respectively. Similar coordination numbers and energetic parameters are also observed for zwitterion association around the existing carboxyl group and are summarised in the supplementary table S3. The key structural observations of CO₂ loaded glycine carbamate solutions made possible through neutron diffraction and structural refinement are also summarised in Table 2.

Finally, we can visualise the structural information presented in Figs. 4 and 5 using 3-dimensional spatial density functions (SDFs). These are isosurfaces containing the most likely positions of a given atom type relative to a central atom type calculated over a given radial distance. This allows us to clearly observe the differences between the cationic associations around both the unloaded glycinate amine group, and the CO₂ loaded glycine carbamate group. These data are shown in Fig. 6 and serve as another good demonstration of the insight that the combined approach of neutron diffraction and computational modelling can offer in the study of carbon capture solutions. The distributions of calculated glycinate amine/glycine carbamate – cation EPSR-derived pair interaction energies from the EPSR simulations are then plotted as box and whisker plots in Fig. 6(c).

### Predicting dominant interactions
The unparalleled access to experimentally constrained atomistic information offered by neutron diffraction and structural refinement

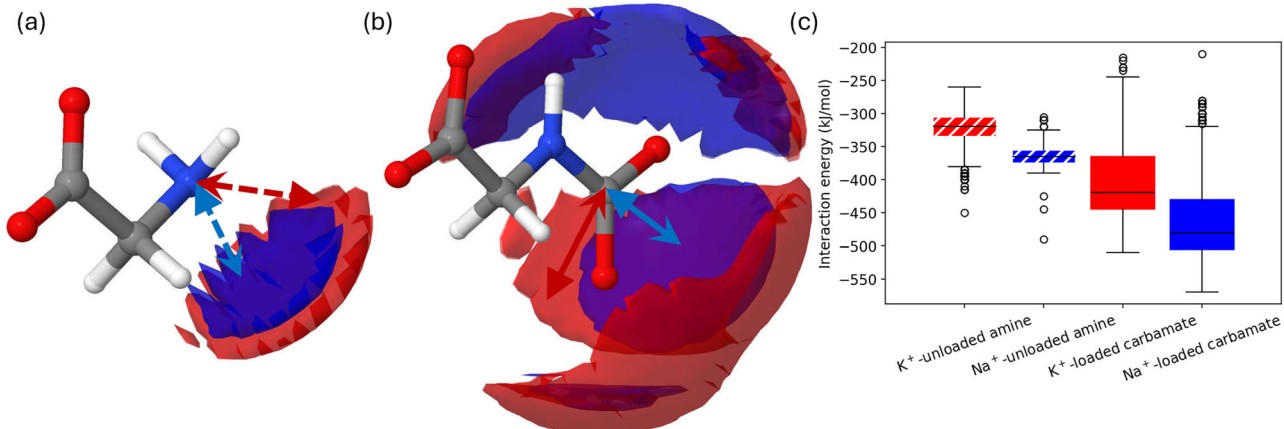

(a) Unloaded: amine – cation (3.16 Å)  (b) Loaded: carbamate – cation (4.20 Å)  (c) EPSR-derived pair interaction energies

**Fig. 6 | Carbamate anion – metal cation interactions.** Spatial density plots (SDFs) depicting isosurfaces for the 50% most probable locations for the K⁺ (red) and Na⁺ (blue) cations around the unloaded glycine amine group (**a**) and the CO₂ loaded glycine carbamate group (**b**) are calculated over the indicated distances. These serve as clear visual demonstrations of the results gained from the RDFs and calculated EPSR-derived pair interaction energy values. The Na⁺ ions reside closer to these chemical groups on the glycinate/glycine carbamate molecules in solution relative to the K⁺ ions, and the association is more well defined, as evidenced by the smaller size of the Na⁺ isosurfaces relative to the K⁺ isosurfaces. These demonstrate the calculated more stable EPSR-derived pair interaction energetic interactions for Na⁺ - glycine compared with K⁺ glycine, which are also shown as box and whisker plots (**c**). As described in the methods section, EPSR-derived pair interaction energies are calculated for appropriate pairs of molecules as determined by their interatomic distances. The disordered nature of the solution state simulations means that this results in a distribution of calculated energies. The coloured boxes show the interquartile range of these distributions for each of the 4 described interactions, with the median indicated by a black line. The whiskers extend to the furthest datapoints which lie within a factor of 1.5 of the interquartile range from the upper and lower quartiles, with any points existing outside this range plotted explicitly as circles.

finally allows us to comment on the likely dominant physical interactions within aqueous carbon capture solutions. In the unloaded state we observe clear structuring of water and cations around the glycinate amine group, as shown in Fig. 4, and around the glycinate carboxylate group shown in the supplementary Fig. S13. While the amine – cation interactions are predicted to be much more energetically stable than the amine – water interactions, due to the absolute charge of the cation, the calculated coordination numbers given in Table 1 show that they occur far less frequently. Simply taking the product of these two values allows us to weigh the influence of the EPSR-derived pair interaction energies and estimate whether amine – water or amine – cation interactions are more dominant in the unloaded state. These calculated values are given in the supplementary table S4. This reveals that amine – water interactions contribute about twice the EPSR-derived pair interaction energetic contribution to the amine coordination shell when weighted by coordination number compared with the amine – cation interactions and should therefore be considered the more dominant. Similarly, we can comment on the coordination number weighted EPSR-derived pair interaction energetic interactions around the loaded carbamate group. The same analysis reveals that in this case the weighted carbamate – water interaction is effectively equal to the weighted carbamate – cation interactions, as the product of their energetic interaction and coordination numbers are effectively equal. However, despite its relatively stable calculated EPSR-derived pair interaction energetic interaction, due to the very low carbamate – glycine zwitterion coordination number (Table 2), the overall weighted energetic contribution of glycine zwitterions to the carbamate coordination shell is effectively negligible. It is worth noting at this point that in amine systems that do not contain stoichiometric K/Na metal ions that amine/carboxylate interactions may be much more significant. Previous neutron diffraction data on aqueous zwitterionic glycine at a molar ratio of 172:5160 glycine:water molecules suggested a $N_{C_{zwit}N_{zwit}}$ coordination number of 0.25[14]. Another neutron diffraction study of aqueous glycine at 1:17 glycine:water molecules observed significant clustering of glycine molecules through hydrogen bonding between their amine and carboxyl groups[55], forming chains of up to 7 molecules.

## Discussion

In this work we have demonstrated the power of neutron diffraction and structural refinement in providing a detailed experimentally constrained, atomically resolved view of aqueous amino acid salts as model amine-based carbon capture solvents before and after CO₂ loading. We have focussed on three main structural features involved in the CO₂ loading and solvent regeneration process through either the two-step or termolecular mechanism previously reported to occur in amine-based solvents and used this information to verify the necessary starting structures for these two routes, as well as quantifying the associated EPSR-derived pair interaction energetic parameters. However, as this technique provides an atomistic view of the system, it grants access to any other structural correlations one may care to investigate, provided that the solutions can be prepared to a suitably high concentration that the interatomic correlations of interest appreciably contribute to the raw scattering data. We predict that in the unloaded state that amine – water interactions are the dominant structural feature, whereas in the CO₂ loaded state that carbamate – water and carbamate – cation interactions are roughly equally dominant. As an additional observation, we have identified underlying structures in solution which can help understand the previously reported[32] and here experimentally verified superior CO₂ loading performance of potassium amino acid salts compared with sodium amino acid salts. While neutron diffraction and structural refinement has been previously applied to a host of aqueous systems[14,20–26], including aqueous electrolytes, amines, and amino acids, this report details an investigation into a carbon capture system in both the unloaded and CO₂ loaded state. As such, there now exists an extremely wide and promising parameter space of various carbon capture solutions to explore with atomic resolution, including blends of complementary agents and non-amine-based solvents, to aid in their intelligent design and large-scale usage.

## Methods

### Materials

NaOH pellets, KOH solution, KOD solution, $D_2O$, and glycine were purchased through Sigma-Aldrich and used without further purification. $D_5$-glycine was purchased from CK isotopes and used without purification. Unloaded Na/K glycinate were prepared gravimetrically by neutralising glycine with equimolar aqueous sodium hydroxide or potassium hydroxide, followed by dilution to the desired concentration with deionized water. A total of 7 H/D isotope variants of unloaded samples were prepared in the same way using the appropriate starting chemical H/D variants, and are listed in supplementary table S5.

### CO₂ loading

5 g aliquots of the unloaded solutions were dispensed into 14 mL glass vials containing a magnetic stirrer and sealed with Suba-seals®. The sample headspace was then evacuated using a needle connected to a vacuum line. Pure $CO_2$ gas was dispensed into capture solution using a gas-tight glass syringe filled with a $CO_2$ volume corresponding to the desired molar loading according to the ideal gas law under constant stirring. The vial headspace was then backfilled with $N_2$ to atmospheric pressure and stirred for a further 20 mins to ensure full absorption of $CO_2$. 100% absorption efficiency by glycinate was assumed. The overall scattering level of the samples measured using Gudrun software[61], which is dependent on the abundance and isotope makeup of the samples[17], was consistent with the expected loading levels. Samples were checked at this stage and by this method to avoid unnecessary exposure to atmosphere resulting in additional $CO_2$ capture and potential precipitation of bicarbonate. In the event of slightly lower $CO_2$ loading than estimated the EPSR fits to the $CO_2$ loaded experimental data would likely be of lower quality and yield increased R factors. However, the low calculated R factors, lower even than the unloaded samples, demonstrate high quality fitting and suggest accurate $CO_2$ loading estimates. Lower loading would likely be most evident in the Fourier transformed raw neutron diffraction data, shown in supplementary Fig. S5, where the intensity at the r value corresponding to the CO and CN bond lengths (-1.3 and 1.4 Å respectively) would be lower in the scattering data compared with the simulated fits, but this is not observed. Lower loading would likely cause the actual coordination numbers associated with glycine carbamate to be lower by virtue of a lesser abundance of this molecular species, but all findings presented in this work would likely be qualitatively correct.

### Vapour liquid equilibrium measurements

VLE measurements were performed on custom built VLE apparatus at C-Capture Ltd., Leeds, UK, at 25 °C. This consisted of a stirred, jacketed stainless steel pressure vessel connected to a $CO_2$ reservoir (burette) containing a known amount of $CO_2$ at an initial pressure of 30 bar through a regulator valve (diagram shown in supplementary Fig. S14). 200 g of Na/K glycinate solution at 2.17 mol/kg $H_2O$ were prepared as previously described and charged into the vessel and sealed. The vessel headspace was then evacuated. $CO_2$ was introduced to the vessel via the regulator valve, allowing the pressure in the vessel to be fixed at 5 bar. This relatively low pressure meant that the $CO_2$ present in the vessel can be modelled using the ideal gas law, reported in Eq. (2), where P is pressure, T is absolute temperature, V is volume of the headspace above the aqueous sample, n is number of mols of $CO_2$, and R is the ideal gas constant.

$$PV = nRT \qquad (2)$$

The high pressure in the burette requires that a modified form of the ideal gas law, the Beattie-Bridgeman equation[62] of state, be employed, as described in Eq. (3), where P is pressure, T is absolute temperature, $\nu$ is molar volume, R is the ideal gas constant, and

$A_0, a, B_0, b,$ and $c$ are empirical gas dependent constants, with the accepted values for $CO_2$ taken as 507.2836, 0.07132, 0.10476, 0.07235, and $6.60 \times 10^5$ respectively[63].

$$P = \frac{RT}{\nu^2}\left(1 - \frac{c}{\nu T^3}\right)(\nu + B) - \frac{A}{\nu^2}$$

$$A = A_0\left(1 - \frac{a}{\nu}\right)$$

$$B = B_0\left(1 - \frac{b}{\nu}\right) \qquad (3)$$

This therefore allows for calculation of the number of moles of $CO_2$ in the burette and the number of moles of unreacted $CO_2$ in the headspace of the vessel, which is in equilibrium with the number of moles of $CO_2$ in the capture solution, as the absorption reaction proceeds. The difference between the sum of these two values and the number of moles of $CO_2$ originally in the burette therefore yields the number of absorbed moles of $CO_2$ by the solution. One can therefore generate time-resolved data and calculate the rate of $CO_2$ uptake by the carbon capture solutions, and associated reaction kinetics. This technique has been well applied to measure reaction kinetics of a host of carbon capture solutions[32,35,64,65].

### Nuclear magnetic resonance spectroscopy

$^{13}C$ NMR spectra were obtained using a two channel Bruker AV-Neo NMR spectrometer operating at 500 MHz equipped with a 5 mm DCH cryoprobe at 298 K. Samples consisted of 600 µL of $CO_2$ loaded solutions mixed with 100 µL $D_2O$ containing 35 mM sodium trimethylsilylpropanesulfonate (DSS) as an internal standard and locking frequency reference. A $^{13}C$-NMR carbon sequence compromising 1600 scans was used to determine the presence of bicarbonate in the loaded glycinate samples. $^{1}H$ $T_1$ relaxation time and diffusion coefficient data were obtained using a Magritek Spinsolve 43 MHz NMR spectrometer and a Bruker Avance II 400 MHz NMR spectrometer respectively. Diffusion coefficient data derived using the principle of pulsed field gradient spin echo and fitting the resultant data to the Stejskal-Tanner expression[20].

### Neutron diffraction

Raw neutron diffraction data on unloaded samples were taken on the NIMROD instrument at ISIS neutron and muon facility[66], and raw neutron diffraction data on $CO_2$-loaded samples were taken on the SANDALS instrument at ISIS neutron and muon facility. While these two instruments have slightly different detector arrays and subsequently different Q ranges (0.02 to 50 Å⁻¹ for NIMROD and 0.1 to 50 Å⁻¹ for SANDALS), previous studies into isotopically varied water have demonstrated that they produce consistent scattering data, with the largest differences occurring at low Q due to the inherent difficulty of correcting for inelasticity effects from hydrogen atoms[67]. On both instruments, high-quality fits following structural refinement were achieved as measured by the R factor, which were determined to be 0.00026, 0.00020, 0.00012, and 0.00011 for unloaded K glycinate, unloaded Na glycinate, $CO_2$-loaded K glycinate, and $CO_2$-loaded Na glycinate, respectively. Samples were loaded into 1 mm path length null coherent scattering TiZr cans, and all measurements were performed at 25 °C. The data was put into an absolute scale using scattering data from a 3 mm thick VNb alloy of known scattering characteristics. Raw data were then corrected for multiple scattering, attenuation, and inelasticity effects using the Gudrun software[61].

### Empirical potential structure refinement

In the Monte Carlo-based structural refinement technique, EPSR[13], simulated boxes of molecules of matching density and solute

concentrations of experimental samples were created. Density at experimental temperature was determined using an Anton Paar 4100 M density meter, and is shown in Supplementary table S6. Solution densities are reported in Table S4. Boxes for the unloaded samples contained 5767 water molecules, 217 $Na^+/K^+$ ions, and 217 glycinate anions. Boxes for $CO_2$-loaded samples contained 5767 water molecules, 217 $Na^+/K^+$ ions, 65 glycinate anions, 76 glycine zwitterions, and 76 glycine carbamate anions. These box sizes were chosen as a compromise between the largest possible box size that would still allow running in a reasonable time. Each atom was described using a reference potential consisting of two Lennard-Jones components, σ and ε, and a charge $q$. The interatomic potential is then determined using the standard Lorentz-Berthelot mixing rules and a Coulomb potential.

The SPC/E model is used for the reference potential for water, which has been previously well used for EPSR simulations containing aqueous glycine[14,55]. The reference potential for $Na^+$ and $K^+$ ions taken from Mancinelli et al. [48], the reference potential for glycine zwitterion taken from previous work from our group[14], and the reference potential for glycinate anion taken from Sweatman et al. [57]. As the authors are unaware of any potential describing glycine carbamate, one was generated using the molecular structure within LigParGen web server[58], and was deemed sensible given the existing parameters for the glycine zwitterion and anion. Full reference parameters are listed in Supplementary Table S2.

The Monte Carlo simulation was allowed to proceed using this reference potential to equilibrate for >200 steps, while calculating what the expected scattering data corresponding to the simulation. A continuously evolving empirical potential derived from the difference between the simulated and experimental scattering data for all isotopic variants simultaneously was added to drive the simulation towards the experimental data. Once a satisfactory match was achieved, the simulation is allowed to proceed and accumulate statistics over >5000 iterations to calculate $g_{αβ}(r)$s.

### EPSR-derived pair interaction energy calculations

EPSR-derived pair interaction energy calculations for particular atomic pairs were performed as previously[25,26]. A molecular trajectory from EPSR was first generated over 1200 iterations for the unloaded samples, and 2500 iterations over the loaded samples. More iterations were accumulated for loaded samples, as the increased variety of glycine-like molecular species (anion, carbamate, and zwitterion as opposed to simply anion) results in each distinct glycine molecule – molecule to be more weakly weighted in the data, and therefore requires more iterations to improve statistical significance. Only one iteration is then considered for further analysis per 100 iterations to ensure that two considered iterations are uncorrelated, improving statistical significance. Hence, for the unloaded data, 13 iterations are considered, and 25 for the $CO_2$-loaded data.

In the simplest case, the potential between two molecules is calculated when two particular atom types $α$ and $β$ form each molecule are within a threshold distance from one another. These threshold distances are derived from the locations of the first minimum in the corresponding $g_{αβ}(r)$. All employed cutoff distances are shown in Supplementary Table S7. Once this condition is satisfied, their total interaction potential is calculated over every pair of atoms in the molecule pair according to the Lennard-Jones and Coulomb potentials described by the reference potential within EPSR. This analysis is done for every appropriate molecule pair in all considered iterations. The results are then binned, and a normal distribution is fit to determine the average EPSR-derived pair interaction energy value with an associated error.

In the case of bulk water – water hydrogen bonding, the 'bulk' water is first identified as those water molecules whose oxygens are not within a given cutoff distance to any other atom in the simulation, with cutoff distances given in table S1. For a given bulk water molecule

acting as a hydrogen bond acceptor, a water molecule is deemed to be acting as a hydrogen bond donor if its oxygen–oxygen distance is within 3.29 Å, and its oxygen–hydrogen distance is within 2.44 Å in all cases, informed by the location of the first minima in the associated $g_{αβ}(r)$s. Their EPSR-derived pair interaction energetic interaction potential is then calculated as described previously.

### Evaluations of robustness of findings

The findings of this work may be influenced by several factors, including but not limited to: the cutoff distances employed for the calculations of EPSR-derived pair interaction energies and coordination numbers, the reference potentials employed, and the number of iterations over which EPSR was averaged. It is therefore important to determine how robust the findings of this work are if one begins to vary these factors. Briefly, this was achieved for the dependence of the EPSR-derived pair interaction energies on their cutoff distances by varying the cutoff distances by ±10%, recalculating all values, and normalising the average relative change in energy to the 10% variation. The sensitivity of the coordination numbers to the employed calculation distance was evaluated by varying the cutoff distances by ±0.06 Å and recalculating all values. This distance was chosen as the coordination number and $g_{αβ}(r)$ data was binned in 0.03 Å increments. Finally, the sensitivity of the produced $g_{αβ}(r)$s to the employed reference potentials was evaluated by choosing 4 different sets of reference potentials, based on combinations of previous literature and LigParGen-generated force fields, and calculating a relative difference value between the 4 produced $g_{αβ}(r)$s using the same form as that used to calculate the R factor given in supplementary equations S1 and S2. The chosen sets of reference potentials used in this report were selected as those that both provided the lowest R factors between the experimental and simulated scattering data following refinement and were more commonly employed in previous EPSR literature. The full results and descriptions (note S1) of these approaches can be found in the supplementary information. Overall, the intermolecular correlations we predict to be dominant by weighting their EPSR-derived pair interaction energies by their coordination numbers, namely the glycine amine/glycine carbamate – water interactions and the glycine amine/glycine carbamate – cation interactions, are deemed to be very robust. The glycine amine/glycine carbamate – water interactions are shown to be insensitive towards the choice of reference potential, but quantitative analysis of the EPSR-derived pair interaction energies is dependent on cutoff distance. However, whether a particular interaction is more or less energetically favourable in K vs Na systems is always unchanged so long as identical cutoffs are used in both instances. The glycine amine/glycine carbamate – cation interactions are shown to be reasonably sensitive to the choice of reference potential, but quantitative analysis of the EPSR-derived pair interaction energies and coordination numbers is shown to be insensitive to cutoff distance, due to the well-defined association shells and favourable interaction energies.

## Data availability

All data generated in this study have been deposited in a publicly available database under accession code https://doi.org/10.5518/1691. All data are available from the corresponding author upon request. All raw neutron diffraction data generated in this study have been deposited in a publicly available database under accession codes https://doi.org/10.5286/ISIS.E.RB2410275 and https://doi.org/10.5286/ISIS.E.RB2220355. Source data are present. Source data are provided with this paper.

## Code availability

Python scripts for additional analysis routines have been deposited in a publicly available database under accession code https://doi.org/10.5518/1691.

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

## Acknowledgements

The project was supported by a grant from the Engineering and Physical Sciences Research Council (EPSRC) (EP/ P02288X/1) and a European Research Council Consolidator Fellowship/UKRI Frontier Research Fellowship for the MESONET project, UKRI EP/X023524/1 to L. Dougan. We are grateful to the University of Leeds and its alumni for a PhD scholarship to support Daniel Sault. We acknowledge the beamtimes at the ISIS neutron and muon facility (RB2410275 and RB2220355). We acknowledge C-Capture Ltd for use of their VLE apparatus.

## Author contributions

H.L. lead writing of the manuscript, gathered neutron diffraction data on unloaded and $CO_2$-loaded samples, performed associated data correction, EPSR analysis, and wrote and performed additional analysis using bespoke analysis routines. D.S. aided in gathering neutron diffraction data on $CO_2$-loaded samples, prepared all $CO_2$-loaded samples, and gathered all NMR and VLE data. T.F.H. was the instrument scientist for the acquisition of unloaded neutron diffraction data on the NIMROD instrument. T.L.H. was the instrument scientist for the acquisition of $CO_2$-loaded neutron diffraction data on the SANDALS instrument. Both T.F.H. and T.L.H. aided in data correction and EPSR analysis. J.W. supervised VLE data acquisition. C.R. helped with the $CO_2$ loading experimental design and the discussion surrounding reaction mechanisms. L.D. is the corresponding author, co-wrote neutron diffraction access proposals, and aided in discussions around neutron diffraction experimental data and analysis. All authors contributed to writing, correcting, and discussing the manuscript.

## Competing interests

The authors declare no competing interests.
