## [Transparent Peer Review file · Nature Communications]

Visualising Reaction Complexes in Amine Based Unloaded and CO₂ Loaded Carbon Capture Solutions

Corresponding Author: Professor Lorna Dougan

Version 1:

Reviewer comments:

Reviewer #1

(Remarks to the Author)

The authors applied neutron diffraction to study the structural features and arrangement of carbon dioxide when absorbed into aqueous solutions of sodium and potassium glycinate. The study is timely, providing insight at the atomic level into the species and energy involved in CO₂ capture in a liquid medium. The methodology and results appear sound. However, one specific aspect needs to be clarified to remove uncertainty regarding the conclusions, as outlined below.

The NionM⁺ interaction appears to play a key role in determining the enthalpic barrier of CO₂ absorption in K glycinate and Na glycinate. However, the role of the "NionM⁺" interaction relative to the "COOionM⁺" interaction is counterintuitive and not immediately clear, including the reference to the -NH₂ of glycinate as Nion since the amine group is neutral (not ionised).

Line 163 & Figure 4: The values of calculated enthalpic interactions of K and Na cations with glycinate are high (-318 ± 1 and -367 ± 1 kJ/mol for K and Na glycinate, respectively), the interactions are indicated as NionM⁺ shown by a green double-headed arrow pointing between the amine group and the metal ion, but what is the contribution of the carboxylic group of the glycinate? The carboxylate anion-metal cation interaction should be prevalent in the assembly of the chemical species in solution. A sodium cation would overwhelmingly interact with the carboxylate anion, not the neutral amine end. The amine lone pair might contribute weakly, but the primary binding site should be the carboxylate, unless the carboxylate is already involved in other, as yet not evident, interactions. Please clarify this point because it underpins the key findings of the manuscript.

Line 358: Was the 100% absorption of CO₂ to achieve the desired loading of carbon dioxide verified experimentally in at least a few instances, or was it entirely assumed? What would be the impact of lower loadings than intended?

Reviewer #2

(Remarks to the Author)

The reported research is really of high interesting and gives a powerful protocol to determine the differences between various carbon capture agents to be used as CO₂ absorbers using neutron scattering. The manuscript is original and well written, the results clearly explained and the discussion of the main experimental findings are properly done with scientific rigor. The data analysis is convincing and the extracted conclusions are well supported by the experiments.

Therefore, I recommend publication of the manuscript after some minor changes:

- In line 136 some terms are repeated ("is shown in shown")
- It is difficult to distinguish the colors used in Fig. 3.

Reviewer #3

(Remarks to the Author)

The manuscript presents a neutron-diffraction/EPSPR study of aqueous potassium and sodium glycinate in unloaded and CO₂-loaded states, augmented by NMR speciation controls and VLE data. The central claims are: (i) refinement of H/D-

contrast neutron-diffraction data provides atomistically resolved insight into solvent reorganization upon carbamate formation; (ii) systematic K⁺ vs Na⁺ differences in hydration and ion pairing; and (iii) an EPSR-based analysis of pair interactions (termed “interaction enthalpy” in the manuscript) to rationalize structure–property trends. The topic is timely for carbon-capture solvent design and leverages a powerful contrast-variation strategy.

The study assembles an ambitious set of isotopic-contrast measurements across both unloaded and CO₂-loaded states and achieves solid experiment–model agreement, lending confidence to the structural inferences. The narrative is clean and focused, tracing how hydration shells, ion pairing, and the water hydrogen-bond network reorganize upon CO₂ uptake. The side-by-side cation comparison (K⁺ vs Na⁺) is particularly valuable, showing consistent qualitative trends that align with expected differences in charge density and ionic size. The deliberate choice to limit bicarbonate formation at moderate loading is sensible and sharpens the analysis, allowing a clearer view of carbamate-centred chemistry without confounding speciation. The dataset is compelling and the study is well motivated. However, ambiguity about neutrality/composition and the interpretation of EPSR-derived pair energies as enthalpies require substantive revision.

Major concerns (major revisions required)

1. Simulation neutrality and composition transparency (loaded state).

As written, the loaded-state EPSR cell is not exactly charge neutral. The carbamate route consumes two glycinate anions per CO₂ and yields one carbamate (–1) plus one zwitterion (0); electroneutrality therefore requires one additional anion per reaction (e.g., OH(–)/OD(–) under basic conditions, or HCO₃(–) if present). For the 76 glycine zwitterions reported (line 408), this implies 76 additional anions to balance 217 K⁺/Na⁺ ions. The confusion likely arises from counting “glycine-derived species” rather than charges; zwitterions are neutral and cannot balance cations. Please make the loading stoichiometry explicit (carbamate + zwitterion balance; remaining glycinate), correct the species list at lines 407–408, and state clearly how neutrality is enforced. If composition is adjusted, all metrics influenced by long-range electrostatics (coordination numbers, pair-interaction energies) must be revisited. This is a technical blocker.

2. “Interaction enthalpy” nomenclature and interpretation.

The reported quantities are EPSR-derived pair interaction energies within chosen cutoffs; they are not thermodynamic enthalpies or free energies. Presenting them as “enthalpy,” and forming an “overall enthalpic contribution” by multiplying with CNs, risks over-interpretation and invites invalid comparisons to calorimetry. Replace “interaction enthalpy” with “EPSR-derived pair interaction energy” (or equivalent) throughout the text/figures/SI; avoid CN×energy language implying a state function; and add a short limitations statement. Change “overall enthalpic contribution” to “coordination-weighted pair-interaction score” at lines 309, 311, and 316.

3. Cutoff/parameter sensitivity and robustness.

Several conclusions hinge on numerical differences in pair energies and coordination numbers (CNs). Quantify robustness to (i) modest changes in radial cutoffs for first-shell definitions and H-bond criteria, and (ii) plausible variations in reference Lennard-Jones/Coulomb parameters. Provide the sensitivity of pair energies and CNs to ±0.05–0.10 Å cutoff changes and reasonable parameter tweaks, summarising qualitative invariants in the main text and placing plots/tables in the SI.

4. Kinetics vs equilibrium.

The manuscript at times implies validation of kinetic claims (e.g., uptake or reaction routes) using VLE, an equilibrium observable. Either confine claims to equilibrium structure–property relationships or add appropriate kinetic data/citations, clearly separating kinetics from equilibrium thermodynamics/structure.

5. Definition of “bulk water.”

The classification of waters as “bulk” vs “first/second shell” underpins several conclusions. Report the fraction of waters labelled “bulk” for each condition and the sensitivity of water–water metrics to the classification criteria, to substantiate statements about water-network reorganization.

6. Instrument/Q-range disclosure and fit quality in the main text.

Different instruments were used across states, and quantitative fit metrics currently sit in the SI. For transparency, add a concise main-text table summarizing instrument, Q-range, contrasts, and fit quality (e.g., R-factors), and briefly note any checks for cross-instrument systematics.

7. Scope of novelty.

Qualify the “first” claim (lines 56, 107, and 339) to the specific scope—neutron-diffraction/EPSR on amino-acid-salt CO₂ solvents across loaded/unloaded states—and distinguish clearly from related neutron/electrolyte/amine literature to avoid overselling novelty.

8. Uncertainty reporting.

CNs are compared across salts and loadings without visible uncertainties. Add error bars (frame bootstrap plus integration-limit variation) wherever CNs are used quantitatively or combined with pair energies.

Version 2:

Reviewer comments:

Reviewer #3

(Remarks to the Author)

The revision addresses my previous concerns in a thorough and satisfactory manner.

I suggest only minor further polishing: slightly soften superlative language (“very robust”, “unprecedented resolution”) and add a one-sentence summary in the Conclusions explicitly stating which interactions dominate in unloaded vs loaded states. Subject to these small edits, I consider the manuscript suitable for publication.

I have also reviewed the authors' responses to the concerns raised by Reviewer #1 and find that those points are adequately addressed in the revised manuscript and supporting information.

We thank the reviewers for the time they have taken in reviewing our manuscript. We have addressed all questions and updated the manuscript and supporting information accordingly. In addition, we have made all the editorial changes requested by the journal, see end of this document.

Reviewer #1 (Remarks to the Author):

The authors applied neutron diffraction to study the structural features and arrangement of carbon dioxide when absorbed into aqueous solutions of sodium and potassium glycinate. The study is timely, providing insight at the atomic level into the species and energy involved in CO₂ capture in a liquid medium. The methodology and results appear sound. However, one specific aspect needs to be clarified to remove uncertainty regarding the conclusions, as outlined below.

The NionM+ interaction appears to play a key role in determining the enthalpic barrier of CO₂ absorption in K glycinate and Na glycinate. However, the role of the "NionM+" interaction relative to the "COOionM+" interaction is counterintuitive and not immediately clear, including the reference to the -NH₂ of glycinate as Nion since the amine group is neutral (not ionised).

We are very grateful for reviewer 1 for their positive comments and for giving us the opportunity to share data regarding the carboxylate – cation interactions. We too consider this data to be interesting and relevant to specialised carbon capture applications, but did not include it in the first iteration of this work as it did not play a direct role in the absorption chemistry. We have now updated the draft to reflect this change, and clarified the points regarding efficiency of sample loading below.

1.1. Line 163 & Figure 4: The values of calculated enthalpic interactions of K and Na cations with glycinate are high (-318 ± 1 and -367 ± 1 kJ/mol for K and Na glycinate, respectively), the interactions are indicated as NionM+ shown by a green double-headed arrow pointing between the amine group and the metal ion, but what is the contribution of the carboxylic group of the glycinate? The carboxylate anion–metal cation interaction should be prevalent in the assembly of the chemical species in solution. A sodium cation would overwhelmingly interact with the carboxylate anion, not the neutral amine end. The amine lone pair might contribute weakly, but the primary binding site should be the carboxylate, unless the carboxylate is already involved in other, as yet not evident, interactions. Please clarify this point because it underpins the key findings of the manuscript.

Following text added and supplementary information updated accordingly.

‘Unsurprisingly, due to the overall negatively charged nature of the carboxylate group on the glycinate anion, one also clearly observes metal cation – carboxylate

interactions. These are both more frequently occurring, with coordination numbers of 0.54 and 0.61 for K glycinate and Na glycinate respectively when calculated over 4.05 Å (representing the average location of the first minima in the $g(\text{CionM}^+)(r)$ s), and more energetically stable, calculated to be -408 ± 2 and -479 ± 7 kJ/mol respectively. While these correlations do not have an appreciable direct impact on the CO_2 absorption reaction for glycinate salts, and hence are not discussed here in detail, for the interested reader these structural results and the $g(\text{CionM}^+)(r)$ s are shown in supplementary figure S9 and table S1.’

Figure S9. Na^+/K^+ - C2a RDFs for unloaded samples. Presence of higher peaks at shorter distances is indicative of stronger ion – water interactions for Na^+ compared with K^+ .

Feature	K glycinate	Na glycinate
Modal nearest neighbour C_{ionM^+} distance (Å)	3.04	2.63
$\text{C}_{\text{N}_{\text{ionM}^+}}(4.05 \text{ Å})$	0.54	0.61
$E_{\text{EPSR},\text{N}_{\text{ionM}^+}}$ (kJ/mol)	-408 ± 2	-479 ± 7

Table S1. Key structural observations in aqueous K glycinate and Na glycinate determined through neutron diffraction and structural refinement. Modal nearest neighbour distances correspond to locations of first peaks in relevant $g_{\alpha\beta}(r)$ s. Errors on reported energies correspond to fitting error of location of gaussian peak to calculated EPSR-derived pair interaction energy distributions.

1.2. Line 358: Was the 100% absorption of CO_2 to achieve the desired loading of carbon dioxide verified experimentally in at least a few instances, or was it entirely assumed? What would be the impact of lower loadings than intended?

Additional text and references added to the methods section to clarify preparation and assumptions around CO_2 loaded samples and how correct loading was validated.

‘Pure CO_2 gas was dispensed into capture solution using a gas-tight glass syringe filled with a CO_2 volume corresponding to the desired molar loading according to

the ideal gas law under constant stirring. The vial headspace was then backfilled with N₂ to atmospheric pressure and stirred for a further 20 mins to ensure full absorption of CO₂. 100% absorption efficiency by glycinate was assumed. The overall scattering level of the samples measured using Gudrun software⁶¹, which is dependent on the abundance and isotope makeup of the samples¹⁷, was consistent with the expected loading levels. Samples were checked at this stage and by this method to avoid unnecessary exposure to atmosphere resulting in additional CO₂ capture and potential precipitation of bicarbonate. In the event of slightly lower CO₂ loading than estimated the EPSR fits to the CO₂ loaded experimental data would likely be of lower quality and yield increased R factors. However, the low calculated R factors, lower even than the unloaded samples, demonstrate high quality fitting and suggest accurate CO₂ loading estimates. Lower loading would likely be most evident in the Fourier transformed raw neutron diffraction data, shown in supplementary Fig. S5, where the intensity at the r value corresponding to the CO and CN bond lengths (~1.3 and 1.4 Å respectively) would be lower in the scattering data compared with the simulated fits, but this is not observed. Lower loading would likely cause the actual coordination numbers associated with glycine carbamate to be lower by virtue of a lesser abundance of this molecular species, but all findings presented in this work would likely be qualitatively correct.'

Reviewer #2 (Remarks to the Author):

The reported research is really of high interesting and gives a powerful protocol to determine the differences between various carbon capture agents to be used as CO₂ absorbers using neutron scattering. The manuscript is original and well written, the results clearly explained and the discussion of the main experimental findings are properly done with scientific rigor. The data analysis is convincing and the extracted conclusions are well supported by the experiments.

Thank you for this positive feedback.

Therefore, I recommend publication of the manuscript after some minor changes:

- 2.1. In line 136 some terms are repeated ("is shown in shown") – **Repetition removed**
- 2.2. It is difficult to distinguish the colors used in Fig. 3. - **The fit lines in figure 3 have now been increased from size 1 to size 2 and improve their presentation.**

Reviewer #3 (Remarks to the Author):

The manuscript presents a neutron-diffraction/EP SR study of aqueous potassium and sodium glycinate in unloaded and CO₂-loaded states, augmented by NMR speciation controls and VLE data. The central claims are: (i) refinement of H/D-contrast neutron-diffraction data provides atomistically resolved insight into solvent reorganization upon carbamate formation; (ii) systematic K⁺ vs Na⁺ differences in hydration and ion pairing; and (iii) an EP SR-based analysis of pair interactions (termed “interaction enthalpy” in the manuscript) to rationalize structure–property trends. The topic is timely for carbon-capture solvent design and leverages a powerful contrast-variation strategy.

The study assembles an ambitious set of isotopic-contrast measurements across both unloaded and CO₂-loaded states and achieves solid experiment–model agreement, lending confidence to the structural inferences. The narrative is clean and focused, tracing how hydration shells, ion pairing, and the water hydrogen-bond network reorganize upon CO₂ uptake. The side-by-side cation comparison (K⁺ vs Na⁺) is particularly valuable, showing consistent qualitative trends that align with expected differences in charge density and ionic size. The deliberate choice to limit bicarbonate formation at moderate loading is sensible and sharpens the analysis, allowing a clearer view of carbamate-centred chemistry without confounding speciation. The dataset is compelling and the study is well motivated. However, ambiguity about neutrality/composition and the interpretation of EP SR-derived pair energies as enthalpies require substantive revision.

Thank you for this feedback and for the opportunity to include further analysis to support our findings.

Major concerns (major revisions required)

3.1. Simulation neutrality and composition transparency (loaded state).

As written, the loaded-state EP SR cell is not exactly charge neutral. The carbamate route consumes two glycinate anions per CO₂ and yields one carbamate (–1) plus one zwitterion (0); electroneutrality therefore requires one additional anion per reaction (e.g., OH[–]/OD[–] under basic conditions, or HCO₃[–] if present). For the 76 glycine zwitterions reported (line 408), this implies 76 additional anions to balance 217 K⁺/Na⁺ ions. The confusion likely arises from counting “glycine-derived species” rather than charges; zwitterions are neutral and cannot balance cations. Please make the loading stoichiometry explicit (carbamate + zwitterion balance; remaining glycinate), correct the species list at lines 407–408, and state clearly how neutrality is enforced. If composition is adjusted, all metrics influenced by long-range electrostatics (coordination numbers, pair-interaction energies) must be revisited. This is a technical blocker.

The simulation is charge neutral. The unloaded solution is populated by equal numbers of glycinate anions (-1) and K/Na cations (+1), hence neutrality. As the absorption of a CO₂ molecules requires two glycinate anions (total charge -2) as shown in figure 1b, and goes on to produce a zwitterion (charge neutral), to conserve charge neutrality the glycine carbamate must have a charge of -2, and is modelled accordingly. We hope this provides clarification.

3.2. “Interaction enthalpy” nomenclature and interpretation.

The reported quantities are EPSR-derived pair interaction energies within chosen cutoffs; they are not thermodynamic enthalpies or free energies. Presenting them as “enthalpy,” and forming an “overall enthalpic contribution” by multiplying with CNs, risks over-interpretation and invites invalid comparisons to calorimetry. Replace “interaction enthalpy” with “EPSR-derived pair interaction energy” (or equivalent) throughout the text/figures/SI; avoid CN×energy language implying a state function; and add a short limitations statement. Change “overall enthalpic contribution” to “coordination-weighted pair-interaction score” at lines 309, 311, and 316. ✓

We have changed the terminology as suggested. We have substituted all instances of enthalpy, enthalpic, and enthalpies for EPSR-derived pair interaction energy, EPSR-derived pair interaction energetic, and EPSR-derived pair interactions energies respectively. As this phrase is now much longer than the simpler use of enthalpy, enthalpic, and enthalpies, in some instances where it has been used very recently or would contribute to poor sentence flow we have just used energy, energetic, and energies. We believe that in the contexts where this approach has been applied it remains clear that we are referring to the EPSR-derived pair interaction energies, and not a free energy.

3.3. Cutoff/parameter sensitivity and robustness.

Several conclusions hinge on numerical differences in pair energies and coordination numbers (CNs). Quantify robustness to (i) modest changes in radial cutoffs for first-shell definitions and H-bond criteria, and (ii) plausible variations in reference Lennard-Jones/Coulomb parameters. Provide the sensitivity of pair energies and CNs to ± 0.05 – 0.10 \AA cutoff changes and reasonable parameter tweaks, summarising qualitative invariants in the main text and placing plots/tables in the SI.

Thanks for the suggestion to include this information. The following section has been added to the methods section, and corresponding tables to the supplementary information.

‘Evaluations of robustness of findings

The findings of this work may be influenced by several factors including but not limited to: the cutoff distances employed for the calculations of EPSR-derived pair

interaction energies and coordination numbers, the reference potentials employed, and the number of iterations over which EPSR was averaged. It is therefore important to determine how robust the findings of this work are if one begins to vary these factors. Briefly, this was achieved for the dependence of the EPSR-derived pair interaction energies on their cutoff distances by varying the cutoff distances by $\pm 10\%$, recalculating all values, and normalising the average relative change in energy to the 10% variation. The sensitivity of the coordination numbers to the employed calculation distance was evaluated by varying the cutoff distances by $\pm 0.06 \text{ \AA}$ and recalculating all values. This distance was chosen as the coordination number and $g_{\alpha\beta}(r)$ data was binned in 0.03 \AA increments. Finally, the sensitivity of the produced $g_{\alpha\beta}(r)$ s to the employed reference potentials was evaluated by choosing 4 different sets of reference potentials, based on combinations of previous literature and LigParGen generated force fields, and calculating a relative difference value between the 4 produced $g_{\alpha\beta}(r)$ s using the same form as that used to calculate the R factor given in supplementary equations S1 and S2. The chosen sets of reference potentials used in this report were selected as the one that both provided the lowest R factors between the experimental and simulated scattering data following refinement, and were more commonly employed in previous EPSR literature. The full results and descriptions (note S1) of these approaches can be found in the supplementary information. Overall, the intermolecular correlations we predict to be dominant by weighting their EPSR-derived pair interaction energies by their coordination numbers, namely the glycine amine/glycine carbamate – water interactions and the glycine amine/glycine carbamate – cation interactions, are deemed to be very robust. The glycine amine/glycine carbamate – water interactions are shown to be insensitive towards the choice of reference potential, but quantitative analysis of the EPSR-derived pair interaction energies is dependent on cutoff distance. However, whether a particular interaction is more or less energetically favourable in K vs Na systems is always unchanged so long as identical cutoffs are used in both instances. The glycine amine/glycine carbamate – cation interactions are shown to be reasonably sensitive to the choice of reference potential, but quantitative analysis of the EPSR derived pair interaction energies and coordination numbers is shown to be insensitive to cutoff distance, due to the well-defined association shells and favourable interaction energies.

Note S1:

To evaluate the dependence of the EPSR-derived pair interaction energies on their cutoff distances we took the approach of varying the cutoff distances by $\pm 10\%$ and recalculating all values. The relative absolute change for the two determined values for each interaction following cutoff distance increase or decrease were then averaged and normalised to the 10% distance variation. This final parameter was assigned a variable δ , and represents how much a particular EPSR-derived pair

interaction energy is likely to vary given a relative change in cutoff distance, e.g. a δ value of 1 means that if one varies the cutoff distance by 10%, the EPSR-derived pair interaction energy will also vary by 10%. This approach demonstrated that the more stable the interaction, such as glycinate/glycine carbamate – metal cation interactions, yield low δ values (~ 0.1), and hence are insensitive findings to cutoff distances, but weaker interactions, such as hydration of the glycinate/glycine carbamate groups, are more variable (~ 1.6). Full results can be found in supplementary table S8. In all instances however the comparison between whether a particular interaction was more stable in the case of loaded/unloaded K glycinate vs loaded/unloaded Na glycinate when using identical cutoff distances were unchanged. In a similar fashion, the sensitivity of the coordination numbers to the employed calculation distance was also evaluated by varying the cutoff distances by ± 0.06 Å and recalculating all values. This distance was chosen as the coordination number and $g_{\alpha\beta}(r)$ data was binned in 0.03 Å increments. More stable interactions, regardless of how infrequent they may be, such as glycinate amine – metal cation interactions, were less variable than less stable interactions. Full results can be found in supplementary tables S9 and S10. We derived an uncertainty for the reported coordination numbers using a bootstrapping approach, where the simulations were allowed to accumulate statistics over short runs (100 iterations), and the relevant coordination numbers were calculated. This was repeated 10 times to derive an average coordination number and an associated standard error. In all instances these errors were unresolvable to 2 decimal places, and as such have not been reported here. The full results of this approach can be found in supplementary tables S9 and S10.

The sensitivity of the final results to the employed reference potentials is important to evaluate. This is likely to be particularly important for samples that contain several different species, or species that exist at relatively low concentration, as these will have lower contributions to the overall experimental scattering data, and therefore structural refinement will be less effective at resolving their interatomic correlations accurately². With this in mind, we employed four different sets of reference potentials for the most complex samples studied here, the CO₂ loaded K/Na glycinate solutions, to evaluate their influence. These were: (1) the ‘base’ reference potentials described in supplementary table S1, derived using a combination of LigParGen³ and previous data⁴⁻⁷ and which gave the lowest R factor following potential refinement, (2) a set where the glycinate anion force field was substituted for a LigParGen generated alternative, (3) a set where the zwitterionic glycine was substituted for a LigParGen generated alternative, (4) a set where the metal cation force field was substituted for that described by Loche *et al*⁸. These forcefields are detailed in supplementary tables S11-13. The water force field was not varied as it is strongly weighted in the data, and hence is the least likely to be

impacted by force field variation, and commonly employed water reference potentials (SPC, SPC/E, TIP3P) are all observed to be very similar⁹, and the carbamate force field was not varied as this was already generated using LigParGen as a sensible alternative is not available. Using these four different reference potential sets, potential refinement was performed as described and the various $g_{\alpha\beta}(r)$ s were produced. For each interaction of interest, the four different $g_{\alpha\beta}(r)$ s were arithmetically averaged to generate an ‘average’ $\bar{g}_{\alpha\beta}(r)$, and their variability quantified using the same approach as is used to determine the ‘R factor’ employed in structural refinement, as described in equation S2, and assigned the variable Σ . Four identical $g_{\alpha\beta}(r)$ s would therefore produce a value of 0. To understand a value corresponding to 4 largely uncorrelated $g_{\alpha\beta}(r)$ s Σ , we deliberately compared four different $g_{\alpha\beta}(r)$ s, one from each of the four reference potential variations for the CO₂ loaded K glycinate data (O_wO_w, K⁺C_{ion}, N_{zwitterion}C_{zwitterion}, H_wN_{carbamate}). These produce a value Σ_0 of 0.0373. The resultant $g_{\alpha\beta}(r)$ s are shown in supplementary figure S15. As expected, the more strongly weighted a particular interatomic correlation is in the experimental scattering data, due to the combined influence of its coherent scattering length and molar abundance, and frequency of interaction (quantified through coordination numbers), the lower the value of Σ . We therefore observe that correlations involving water are robust ($\Sigma \sim 0.0005$), however correlations involving glycinate/glycine carbamate, and metal cations are more variable ($\Sigma \sim 0.02$), particularly those that are infrequently occurring, such as glycine carbamate – glycine zwitterion interactions ($\Sigma \sim 0.035$). The full results are detailed supplementary table S14. Overall, considering the intermolecular correlations we predict to be dominant by weighting their EPSR-derived pair interaction energies by their coordination numbers, the findings of this work are deemed to be robust. It is important to note at this point that the results deriving from any of these forcefields should not be considered incorrect, as all are sensible starting parameters, merely that for some correlations EPSR is less able to offer a single unique solution than others.’

3.4. Kinetics vs equilibrium.

The manuscript at times implies validation of kinetic claims (e.g., uptake or reaction routes) using VLE, an equilibrium observable. Either confine claims to equilibrium structure–property relationships or add appropriate kinetic data/citations, clearly separating kinetics from equilibrium thermodynamics/structure.

We agree with reviewer 3 that VLE is by definition an equilibrium observable, however the data acquired in this study using this principal monitors time resolved CO₂ uptake by the amino acid salt solutions, as shown in supplementary figure S8,

and therefore does indeed give kinetic information. We have updated the methods section accordingly to highlight this, and added additional references.

‘The difference between the sum of these two values and the number of moles of CO₂ originally in the burette therefore yields the number of absorbed moles of CO₂ by the solution. One can therefore generate time-resolved data and calculate the rate of CO₂ uptake by the carbon capture solutions, and associated reaction kinetics. This technique has been well applied to measure reaction kinetics of a host of carbon capture solutions^{32,35,64,65}.’

We also agree that these two methods are not directly comparable and have updated the manuscript accordingly.

‘We also validate these previous kinetic studies from the literature using time-resolved vapour liquid equilibrium (VLE) measurements shown in supplementary Fig. S8. It is important at this point to note that EPSR provides an ensemble averaged equilibrium structure, rather than time resolved CO₂ uptake data derived through VLE, hence while the two effects are certainly related and logical consistencies clearly exist, one cannot serve as a validation of the other.’

‘These values both lie within error of one another and are slightly less stable than water-water hydrogen bonds found in pure water, previously calculated to be -17.71 ± 0.08 kJ/mol²⁵, suggesting that proton hopping in amine-based capture agents is likely to be slower than in pure water⁴⁴.’ (reference added)

3.5. Definition of “bulk water.”

The classification of waters as “bulk” vs “first/second shell” underpins several conclusions. Report the fraction of waters labelled “bulk” for each condition and the sensitivity of water–water metrics to the classification criteria, to substantiate statements about water-network reorganization.

This is important point raised by reviewer 3. We have now stated clearly what proportion of water molecules were considered ‘bulk’ in the data analysis.

Now added to main text:

‘The bulk water-water hydrogen bond EPSR-derived pair interaction energies, calculated as described previously^{20,25,26} and in the methods section, are measured as -16.9 ± 0.1 and -16.76 ± 0.09 kJ/mol for K and Na glycinate respectively. Using the described cutoff distances means that approximately 24.2% and 25.0% of the total number of available water molecules in the simulations for K-glycinate and Na-glycinate are ‘bulk’ water molecules respectively.’

‘Unlike previously, the K glycine carbamate bulk water network is observed to be less stable than the Na glycine carbamate, with calculated water-water hydrogen

bond EPSR-derived pair interaction energies of -16.04 ± 0.09 and -16.17 ± 0.08 kJ/mol respectively, however these two values lie within error of one another. Using the described cutoff distances means approximately 26.8% of the total number of available water molecules in the simulations for both loaded K-glycinate and Na-glycinate are ‘bulk’ water molecules.’

3.6. Instrument/Q-range disclosure and fit quality in the main text.

Different instruments were used across states, and quantitative fit metrics currently sit in the SI. For transparency, add a concise main-text table summarizing instrument, Q-range, contrasts, and fit quality (e.g., R-factors), and briefly note any checks for cross-instrument systematics.

Differences between the two instruments are now more clearly described in the main text, along with previous literature providing a direct comparison between the two. We also quote the R factors explicitly.

‘While these two instruments have slightly different detector arrays and subsequently different Q ranges (0.02 to 50 \AA^{-1} for NIMROD and 0.1 to 50 \AA^{-1} for SANDALS), previous studies into isotopically varied water have demonstrated that they produce consistent scattering data, with the largest differences occurring at low Q due to the inherent difficulty of correcting for inelasticity effects from hydrogen atoms⁶⁵. On both instruments high quality fits following structural refinement were achieved as measured by the R factor which were determined to be 0.00026 , 0.00020 , 0.00012 , and 0.00011 for unloaded K glycinate, unloaded Na glycinate, CO₂ loaded K glycinate, and CO₂ loaded Na glycinate respectively.’

3.7. Scope of novelty.

Qualify the “first” claim (lines 56, 107, and 339) to the specific scope—neutron-diffraction/EP SR on amino-acid-salt CO₂ solvents across loaded/unloaded states—and distinguish clearly from related neutron/electrolyte/amine literature to avoid overselling novelty.

We have updated the manuscript to clarify.

‘In this work we complete the first study of an amino acid salt based carbon capture solvent in both the unloaded and CO₂ loaded state using neutron diffraction’

‘These structures are outlined in Fig. 2 and numbered in relation to the interactions above. For the first time, neutron diffraction is used to examine these crucial structural features and quantify the energetic interactions between species

present in amine-based carbon capture solutions, amino acid salts in this case, before and after CO₂ loading at unprecedented resolution.’

‘While neutron diffraction and structural refinement has been previously applied to a host of aqueous systems^{14,20–26}, including aqueous electrolytes, amines, and amino acids, this is the first report of a detailed investigation into a carbon capture system in both the unloaded and CO₂ loaded state.’

3.8. Uncertainty reporting.

CNs are compared across salts and loadings without visible uncertainties. Add error bars (frame bootstrap plus integration-limit variation) wherever CNs are used quantitatively or combined with pair energies.

Please see comment 3.3

Editorial changes:

Source data included

- All RDFs (.g01 files)
- All CNs (.z01 files)
- Raw scattering data (.t01 files)
- Scattering fits (.u01 files)
- Interaction energy bins

** Please replace your bar graphs with plots that feature information about the distribution of the underlying data. All data points should be shown for plots with a sample size less than 10. For larger sample sizes, please consider box-and-whisker or violin plots as alternatives. Measures of centrality, dispersion and/or error bars should be plotted and described in the figure legend.*

Bar graphs in new figure 6 replaced with box and whisker plots with additional information in figure caption and in the main text:

Additional main text: The distributions of calculated glycinate amine/glycine carbamate – cation EPSR-derived pair interaction energies from the EPSR simulations are then plotted as box and whisker plots in Fig. 6(c).

Additional figure caption text: ‘These demonstrate the calculated more stable EPSR-derived pair interaction energetic interactions for Na⁺ - glycine compared with K⁺ glycine, which are also shown as box and whisker plots (c). As described in

the methods section, EPSR-derived pair interaction energies are calculated for appropriate pairs of molecules as determined by their interatomic distances. The disordered nature of the solution state simulations means that this results in a distribution of calculated energies. The coloured boxes show the interquartile range of these distributions for each of the 4 described interactions, with the median indicated by a black line. The whiskers extend to the furthest datapoints which lie within a factor of 1.5 of the interquartile range from the upper and lower quartiles, with any points existing outside this range plotted explicitly as circles.'